# Sex Differences in Brain Disorders

**DOI:** 10.3390/ijms241914571

**Published:** 2023-09-26

**Authors:** Malgorzata Ziemka-Nalecz, Paulina Pawelec, Karolina Ziabska, Teresa Zalewska

**Affiliations:** NeuroRepair Department, Mossakowski Medical Research Institute, Polish Academy of Sciences, 5, A. Pawinskiego Str., 02-106 Warsaw, Poland; mnalecz@imdik.pan.pl (M.Z.-N.); ppawelec@imdik.pan.pl (P.P.); kziabska@imdik.pan.pl (K.Z.)

**Keywords:** sexual dimorphism, neuropsychiatric disorders, autism spectrum disorder (ASD), Down syndrome (DS), Rett syndrome (RTT), fragile X syndrome (FXS), bipolar disorder (BD), epilepsy, schizophrenia (SZ), major depressive disorder (MDD), post-traumatic stress disorder (PTSD)

## Abstract

A remarkable feature of the brain is its sexual dimorphism. Sexual dimorphism in brain structure and function is associated with clinical implications documented previously in healthy individuals but also in those who suffer from various brain disorders. Sex-based differences concerning some features such as the risk, prevalence, age of onset, and symptomatology have been confirmed in a range of neurological and neuropsychiatric diseases. The mechanisms responsible for the establishment of sex-based differences between men and women are not fully understood. The present paper provides up-to-date data on sex-related dissimilarities observed in brain disorders and highlights the most relevant features that differ between males and females. The topic is very important as the recognition of disparities between the sexes might allow for the identification of therapeutic targets and pharmacological approaches for intractable neurological and neuropsychiatric disorders.

## 1. Introduction

A remarkable feature of the brain is its sexual dimorphism. Brain dimorphism is a complex process, with multiple contributing mechanisms and pathways resulting in differences. Sex-based differences in terms of prevalence, age of onset, pathophysiology, and symptomatology have been identified in a range of neuropsychiatric illnesses (Figure 1). Mechanisms determining sexual dimorphism in pathological states may vary according to location and species. The majority of the literature indicates the key role of a complex interaction between genetic factors and sex hormones, which are probably responsible for the different susceptibility of men and women and may change the severity of symptoms. However, other sex-specific factors, e.g., environmental factors and gene–environment interactions, are also considered. It should be noted that many of the participating factors suggested by researchers remain elusive and require further attention. Despite recent advances, the precise link between the proposed agents and brain impairments is not completely understood. The present paper aims to provide an overview providing up-to-date information on the more recent findings on sex-related differences in neuropsychiatric diseases (Table 1). A better understanding of the mechanisms underlying sex-based answers to pathological conditions may be important in the identification of therapeutic targets and pharmacological treatments for these intractable disorders. 

## 2. Autism Spectrum Disorder (ASD) 

Autism spectrum disorder (ASD) comprises a heterogeneous group of neurodevelopmental disorders defined by the early developmental onset of continuous life-long symptoms characterized by social communication deficits and a pattern of restricted behaviors of interest [50,51]. One of the most widely accepted theories is that autism stems from some neurodevelopmental anomaly (genetic, environmental, and hormonal). Since the etiopathogenesis of autism has not been elucidated so far, the diagnosis of individuals is mainly based on the presentation of three symptoms: profound alteration in social interaction, communication deficits, and stereotyped behaviors. A comparative clinical study indicated a significant predominance of ASD pathology in males. According to available information, males are about 3–4 times more likely to be diagnosed than females [6,7,8,9]. So far, some contrasting data postulate that females have less restricted interests and activities and more significant social communication difficulties than males with ASD [3,4,5]. In contrast, one study reported that females and males with ASD did not differ in terms of the severity of autism [52]. Thus, till now, the results concerning differential sex effects in ASD are highly variable. The inconsistency of subtle yet potentially meaningful ASD phenotypic differences between males and females results from methodological issues [53]. It should be mentioned that while there is no evidence of sex differences in early childhood cognitive functioning, an executive functioning difference exists in adulthood. Therefore, new research will probably be needed for the elucidation of the differences between sexes, covering both the onset and severity of the symptoms. 

In contrast to the clinical presentation, only a few studies have systematically explored the effect of sex in mouse models of autism. A summary of the knowledge about sex-dependent behavioral differences is provided by Romano et al. [54]. However, because of the prevalence of males in the analyzed group, it was rather impossible to draw any conclusion. Recent data in models suggest that females with autism when compared to affected males, present less severe repetitive/restricted behaviors but comparable deficits in social and communication domains [55]. 

The precise mechanism responsible for the differences between men and women remains unclear. Many researchers considering the developmental effects of steroid hormones in the brain during premature organization, as well as the maintenance of sexually dimorphic structures, have focused attention towards their engagement in setting sex-specific liability [7,56,57]. The action of hormones was further strongly supported by the finding that abnormal exposure to testosterone during pregnancy correlated positively with several autistic features and inversely with social skills in children and had a profound effect on sexual dimorphism in human behavior and ASD [1,2]. Testosterone may interact with downstream molecules like neurotransmitters, neuropeptides, and immune pathways and, by these, contribute to male vulnerability. In an interesting report, Baruch et al. (2018) state that altered androgen and estrogen levels could both lead, via different pathways, to alterations of cognitive, linguistic, social and emotional circuits [57]. 

Male and female hormones control the expression of autistic candidate genes differently. For example, the candidate gene for the increased susceptibility to autism is the *RORA* gene, the expression of which is low in the frontal cortex of autistic individuals and oppositely regulated by the male and female hormones. Males seem to be more susceptible to disruptions in RORA and an increase in testosterone [58]. Therefore, genes associated with ASD may be modulated by sex hormones.

Various studies have attributed the male preponderance of ASD to sex-specific single-nucleotide polymorphism, single-nucleotide variants, copy number variants, and proteins (some of them involved in synapse formation and cell adhesion [8,59,60,61,62]. However, the above findings have not been consistently replicated [63]. 

Despite several advances, the precise link between the considered factors and sex-based differences in ASD is not completely understood. The summary of existing hypotheses and phenomena that contribute to the differences between males and females in ASD are presented in an elegant, detailed review prepared by Ferri et al. [64]. This work may provoke researchers to perform further studies for the recognition of sex-based mechanisms in ASD, which are necessary for the introduction of more specific therapies.

## 3. Down Syndrome

Down syndrome (DS) is the most common genetic cause of intellectual disability, with a prevalence of 1 in every 700 births. Down syndrome (DS) is caused by partial or full triplication of human chromosome 21 (Hsa21) [65]. Data taken from the literature indicate a prevalence of males in children with all trisomy 21 variants with a sex ratio of 1.24. The most prominent feature of DS is the learning and memory deficiency that affects 100% of patients. The disease also demonstrates other behavioral phenotypes, such as the incomplete and delayed acquisition of motor, linguistic, and visual–spatial abilities and neurobehavioral disorders. It was detected that DS males, as compared to females, have major behavioral problems, including attention deficit, hyperactivity-like behaviors in childhood, and depression in adulthood [12]. Other occurring phenotypes diagnosed in DS patients are related to cardiovascular disturbances, breathing, and skeletal alterations. It was noted that males were affected twice as much as females, suffering a higher prevalence of breathing disorders during sleep [13]. In contrast, a lower bone mineral density was detected in DS females [66]. 

Research examining sex differences postulated a general agreement that women with DS are at higher risk of developing Alzheimer-like dementia [10,11]. The early age of menopause onset in relation to DS and the loss, as a consequence, of endogenous bioavailable estradiol levels cause women to be four times more likely to develop Alzheimer disease (AD) [67]. Thus, levels of estradiol were found to correlate with the onset of cognitive decline in female DS patients [68]. It is known that the hormonal status of DS patients appears to be highly affected, but to date, no clinical trials concerning estrogen, as well as other hormone replacement therapies, have been published for women with DS. In contrast to the lack of clinical trials in humans, a prolonged estrogen treatment improved cognitive deficits in Ts65Dn mice and restored deficits in the brain’s cholinergic system and function [68,69,70,71]. The results might be coupled to the normalization of the serum levels of estrogen, which was reduced in DS female mutant mice. This leads us to the conclusion that estrogen probably plays a key role in delaying the onset and severity of dementia in women with AD [72,73]. Thus, the results of the preclinical study forced us to assess this speculation in human patients.

## 4. Rett Syndrome 

Rett Syndrome (RTT) is a progressive X-linked neurodevelopmental disorder. RTT affects almost exclusively girls, with a prevalence of 1 in 10,000 births [15,16,17,18]. Diagnosis of RTT in males is rare as patients do not often survive past infancy [19], except for a few cases recognized beyond early childhood [74,75,76]. Typical symptoms are onset at months 6–18 after normal neurological and physical development and progress over several stages with a regression period characterized by a profound loss of acquired developmental communication skills [77]. Further on, disease development reaches a plateau, joined by deterioration of motor abilities, breathing and feeding disturbances, EEG impairment, sleep problems, and autistic-like behaviors [14,77,78,79]. Rett Syndrome is the most abundant cause of intellectual disability, neurological regression, loss of speech, and learning deficits. One of the most common symptoms of RTT is epilepsy, which occurs in 60–80% of RTT cases. Neuropathological studies showed that RTT patients exhibit reduced brain volume and neuronal hypotrophy [14]. The studies have reported that RTT patients also demonstrate other symptoms caused by metabolic complications in organs other than the brain [80,81]. These include gastric problems, cardiorespiratory problems, decreased bone density, dyslipidemia, inflammation of the gallbladder, urological dysfunction, and sleep disturbances [79,82]. 

RTT results primarily from mutations in the X-linked gene (*MECP2*), encoding methyl-CpG binding protein 2 (MeCP2) [83]. The protein MeCP2 binds to methylated genomic DNA and functions as a transcriptional regulator through interactions with other important genes [84]. Evidence indicates that MeCP2 expression correlates with the postnatal maturation of the CNS and neuronal differentiation [85]. Importantly, MeCP2 plays a particularly significant role in the development and maintenance of functional mature neurons [86,87]. Mutation in MeCP2 accounts for more than 90% of classical RTT causes [14]. Moreover, mutation status combined with a degree of X chromosome inactivation and the presence of modifier genes might be a predictor of disease severity [79]. Furthermore, selectively re-expressing MeCP2 in the postmitotic neurons of adult mice results in partially reversing the symptoms of RTT [88,89,90]. 

Clinical presentation in Rett Syndrome differs significantly among individuals based on the degree of phenotypic variation resulting from specific mutations. 

The monogenic nature of the disease has led to the development of Mecp21-mutant mouse models, some of which recapitulate certain phenotypes of human RTT. A comparison of behavioral analyses showed sex-dependent differences in the major mouse model of RTT [54]. The effects could be linked to differential expression of MeCP2 in male and female brains during development [91]. It seems logical to hypothesize that in the early programming of sex-based differences, a complex interaction between sex hormones and MeCP2 takes place. Unfortunately, in contrast to the reported data indicating the involvement of hormones in neurodevelopmental diseases, there is no information on the levels of sex hormones at different ages in RTT mouse models. Further knowledge on this topic may be necessary for a better understanding of the different roles played by MeCP2 in the development of sex-based differences [54]. 

The studies in the RTT field have recently focused on finding an effective treatment. One of the considered strategies is to deliver a functional version of the gene to increase MeCP2 expression and could compensate for the loss of function [92]. The other one is to target downstream factors of MeCP2 by improving neurotransmitter signaling or upregulation genes under MeCP2 control. 

Introducing the first strategy may have functional implications, as MeCP2 overexpression may reactivate X-linked genes to pathogenic levels. Thus, it is necessary to use a molecule that can reactivate only MeCP2 expression [93,94].

The second strategy requires the identification of the precise functions of MeCP2, which are currently still limited [79]. Therefore, future research should consider improving the quality of life of RTT patients.

## 5. Fragile X Syndrome

Fragile X Syndrome (FXS) is a monogenetic neurodevelopmental disorder caused by an expansion of unstable trinucleotide repeats (CGG) within the promoter of mental retardation gene 1 (*FMR1*), encoding for the Fragile X Mental Retardation Protein (FMRP) [21,95,96]. FMRP is an RNA-binding protein involved in several functions through the modulation of the translation of several proteins engaged in brain activity. Additionally, FMRP, as a component of the ribonucleoprotein complex, participates in shuttling between the nucleus and cytoplasm, in transport along dendrites, and in association with polyribosomes [22,97,98,99]. Genetic aberration reduces or even eliminates FMRP, leading to neurodevelopmental impairments. Studies have shown that the lowered level of FMRP expression correlates with the degree of disorders [100,101]. Unexpectedly, the assessment of function suggests that despite the higher level of FMRP in females vs. males, females’ FXS do not necessarily achieve better outcomes than their male counterparts [102]. 

Clinical features of individuals with FXS include intellectual impairment and stereotyped behaviors such as locomotor hyperactivity, sensory hypersensitivity, and unusual physical marks such as an elongated face, flat feet, and extended finger joints. Fragile X Syndrome is usually diagnosed around 2–3 years of age and predicts life-long disability. Till now, disease-modifying drugs are still not available [103,104]. The data sets present a lower incidence of FXS in females, at a ratio of 1:7000, while in males, this FXS syndrome affects 1 in every 4000 of the population [105,106]. In general, female human patients typically display less severe symptoms than males, probably due to compensation by the second not-affected X chromosome [20]. In contrast, others have thought that such compensation alone may not be sufficient to explain all sex-dependent differences. 

Recent data postulated the regulation of *Fmr1* gene expression by sex hormones, particularly 17 beta-estradiol and testosterone [54]. The precise analysis performed by Yang et al. [107] found that the alteration in estradiol level in Fmr1 KO neurons is associated with changes in LTP, synaptic architecture, and fear memory. This finding supports a complicated interaction between FMRP and sex hormones, although the research has addressed that other factors, such as different development or different neural circuitry, also participate in these processes [108]. 

Sex-dependent differences have been evidenced in cognitive and clinical profiles [109]. Investigations have challenged the notion that FXS males present a greater likelihood of displaying autistic-like behaviors, while females may have normal intelligence or mild intellectual disability [21,22,23]. 

The topic of sex-dependent differences in symptoms in FXS patients was developed further in preclinical rodent models in rats [110] and mice [21,111,112]. The data obtained via an analysis of Fmr1 KO of both sexes are consistent with the above-described milder phenotype in females compared to affected males [113,114,115,116]. The detailed study of Wong’s research group found differences in EEG and in repetitive behavior tests, wherein males displayed increased activity while females did not. In addition, females but not males showed abnormal fear memory. In the opinion of Nolan et al. [117], the effect is probably associated with the loss of FMRP.

However, there is a lack of information concerning the interaction between the investigated genotypes and hormones in Fmr1 KO animals. So far, there are also no data presenting hormone profiles in these models. Therefore, the discussion about the potential therapeutic value of the pharmacological modulation of hormone levels is, at present, impossible and requires more studies.

## 6. CDKL-5 Associated Disorders (CDD)

Cyclin-dependent kinase 5 (CDKL-5 known as STk9) encodes a serine/threonine kinase, with the highest levels of expression in the brain [118]. CDKL-5 protein is vital for normal brain development and function. It participates in basic processes such as neuronal morphogenesis and plasticity through interaction or association with other proteins or via phosphorylation modification [119,120,121,122]. A lack of CDKL-5 expression results in decreased dendritic branching, impaired neuronal circuit connection, impaired synaptic vesicle stability, and neurite outgrowth, as demonstrated in vitro [123,124,125] and in vivo [120,122,126].

Mutations in the X-linked *CDKL5* gene and consequential ion deficiency in cyclin-dependent kinase-like 5 caused severe neurodevelopment disorders (CDD). Individuals suffer infantile-onset drug-resistant seizures and lifelong disability, including motor, visual and social impairments. CDKL-5 disorders primarily affect females despite CDD being rather rare, with a prevalence of 1:40,000 or 1:60,000 births [127,128,129]. Pathogenic variants in CDKL-5 are among the most common genetic causes of severe epilepsy in childhood. No effective treatment is currently available, and medical management is only symptomatic and supportive.

The generation of knock-out mice carrying inactivating mutations in the catalytic domain allowed for the characterization of the in vivo consequences of the loss of function of this protein [130,131,132]. Behavioral research shows that mutant mice display impulsivity together with defective cognitive performance and autistic-like abnormalities. However, there are no complete data comparing the phenotype of males and females in this KO model. Only one study found a sex-dependent subtle difference in locomotor activity [130]. However, data are very scarce and do not allow for precise conclusions. 

## 7. Bipolar Disorder

Bipolar disorder (BD) is a serious and insistent complex psychiatric disorder resulting in substantial morbidity and mortality. BD is characterized by extreme fluctuations in mood, demonstrated by manic, hypomanic and depressive episodes. A substantial proportion of patients with bipolar disorder experience neurocognitive impairments [133,134,135]. Lifetime prevalence reaches 1.3–1.6% in all human populations. BD begins in late adolescence or early adulthood with a mean onset age of before 21 years [136]. The majority of studies indicate an equal age of onset across the sexes, although some data have shown that women are slightly older than men when the disease manifests [28,137]. The ratio of BD incidence in men and women is approximately 1:1. Important but still limited and sometimes contrasting findings indicate the existence of sex differences concerning the epidemiological, clinical and psychopathological characteristics of the disease [138]. Usually, women show a higher rate of rapid cycles [27], mixed features [139], a higher number of depressive episodes [28,29] and suicide attempts [30,31]. Conversely, manic episodes and unipolar mania are more common among men [28].

A substantial proportion of patients with BD Ie neurocognitive impairments [135], but this subject has not been explored in the literature relating to BD. Lastly, the field of BD research has focused on the relative contribution of sex to cognitive profiles in BD [140,141]. 

It was found that men with BD perform worse than women in some aspects, like visuospatial construction [142,143], verbal memory performance [144], and reduced sensorimotor gating [145]. By contrast, males performed better in spatial memory and sustained attention tasks [146]. Recently, Xu observed that scores regarding attention and delayed memory were lower in males than in females [140]. In addition, one study showed that women with BD experience greater cognitive benefits from vigorous physical activity than men with BD [147]. 

Previous studies stated that the sex-specific degree of neurocognitive impairments is closely linked to sex dimorphism in brain structures. In truth, no full anatomical analysis was performed on a brain with BD. Only two reports have indicated significant evidence of anatomical differences. Shi [148] observed that right hippocampal volume loss was more evident in females than males. The second publication by Mitchell et al. [149] found that two structures (the left supramarginal gyrus and right inferior parietal lobule) are smaller in males with BD than in females and were correlated with worse cognitive performances. Due to the limited findings, it is still not clear if such differences influence the described cognitive reactions.

Further systematic studies showed that sex-mediated differences in symptoms and cognitive dysfunction in BD are attributed to sex hormones (gonadal and neurosteroids) [150,151,152]. Worsening BD symptoms in women during periods of low levels of sex hormones and in clinical trials indicate that steroid hormones can improve specific symptoms of BD, which is consistent with the above suggestion. Importantly, two other studies showed the benefits of using synthetic progesterone for the treatment of acute mania [153,154]. 

Despite these important findings, speculation that hormones determine cognitive sex differences in BD would be rather premature. The existing literature containing research data showing hormones profile in the BD disorder is still sparse and inconsistent [141]. It seems clear that sex-mediated differences need to be determined in future investigations. 

Apart from hormones, it is thought that genes are likely to explain sex-mediated differences in neurocognitive impairments. However, no distinct gene has been unequivocally associated with BD [24]. Detailed analysis of gene expression in post-mortem tissue has shown dysregulation of several genes and pathways, including those associated with the ubiquitin pathway, synaptic functions [24], intracellular protein trafficking, and protein metabolic processes [25] in BD patients. The reported results are not precise, and future studies will need to address this subject to achieve an understanding of the complex mechanism. 

Furthermore, others have focused on the potential effect of inflammation. A body of evidence implies that the difference in exacerbated inflammation occurring in BD may be a causative factor in different levels of cognitive decline in females and males. Dysregulation of cytokine levels in the peripheral blood and central nervous system (CNS) of patients and the relapsing–remitting nature of the disease might point out the significance of immune responses in the pathogenesis of BD [155]. It has also been detected that an increase in proinflammatory markers TNFalfa and IL8 is associated with both depressive and maniac episodes [26,156,157]. In general, little is known about the potential role of inflammation in sex-based differences. Importantly, only one study reported the upregulated expression of *SOCS* genes (suppressors of cytokine signaling), and only in male subjects [158]. These results provide support to the hypothesis that sex-dependent differences in patients with BD might be, at least partially, explained by inflammation. A significant effect of sex was later confirmed when considering altered inflammation markers in BD phases [159]. More work is needed to prove a direct causal link between inflammation and sex differences.

## 8. Epilepsy

Epilepsy is a chronic neurological disorder that affects more than 70 million people worldwide [160]. The disease is characterized by an enduring predisposition to recurrent spontaneous and unpredictable seizures, which can result in brain damage and substantial body injury. The clinical manifestation of epilepsy includes sudden and abnormal episodes of motor, sensory, autonomic, or psychological origin. The disorder can arise as a result of a dysfunction of the normal connectivity of neural networks or/and imbalance of inhibitory and excitatory neurotransmitters. Comparative clinical data have shown that the overall incidence of epilepsy, the susceptibility to excitability episodes, and the occurrence of epileptic seizures are generally higher in males [32,33,34,35]. Nevertheless, exacerbation of cyclic series seizures in women during the specific phases of the menstrual cycle was also documented. Such occurring episodes are named catamenial epilepsy. Studies indicate that the menstrual-related fluctuations of seizures may be linked to alterations in the concentration of neurosteroids in the brain [161,162,163].

Despite such inconsistency, sex-dependent differences are observed across the different forms of epilepsy and seizure conditions [164]. The mechanisms that cause the establishment of dimorphism in epilepsy still remain unclear. Two groups of researchers reported that sex-linked disparities in regional brain morphology, cerebral asymmetry, and variations in functional connectivity in the brain may determine sex-dependent vulnerability to seizures and epileptogenic cascades [165,166]. However, the precise molecular events that lead from the alteration of brain morphological structures to sexual differences are not completely understood.

A few but convincing studies showed a strong connection between sex hormones and epileptic conditions. It was found that steroids derived from the peripheral system, as well as neurosteroids, likely exert a changeable and complex effect on seizure sensitivity. Since then, these agents have received considerable attention, spurring justified hope that many hormones may represent new therapeutic strategies against seizures [167,168,169,170]. Detailed analysis of neurosteroid (allopregnanolone and androstanediol) function revealed that they act as positive modulators of inhibitory GABAA receptors [171]. Consistent with this, chloride influx into the cell increased, producing neuronal inhibition [167].

Despite the fact that, to date, there have been no studies to demonstrate that reduced levels of neurosteroids can influence epileptogenesis, its effect on neuronal inhibition may serve as a basis for sex differences in epilepsy. One particularly illuminating experiment conducted by Reddy et al. [172] showed that extrasynaptic receptor GABAA contributes to sex differences in seizure susceptibility and neurosteroid protection. In addition, his results confirmed greater sensitivity in females than males. This corresponds partially with the earlier statement that steroid hormones play a key role in the neuroendocrine control of neuronal excitability and seizure susceptibility [173].

It has been shown that the action of another intensively analyzed hormone, progesterone, is associated with anticonvulsant activity due to the negative impacts of glutamatergic transmission [174,175]. It is worth noting that only a few studies performed clinical experiments with progesterone administration. This leads to reduced seizures in women with epilepsy [162,176]. In contrast, estrogen presents a highly variable effect on seizure susceptibility, while, in general, it exhibits pro-convulsant and epileptogenic properties in animals and humans via the potentiation of glutamatergic transmission [174,175,177]; on the other hand, some studies demonstrate protective anticonvulsant activity. It is apparent that the final action is likely due to treatment duration, dosage, hormonal status, and the seizure model [178,179].

Interestingly, several studies have also shown that androgens and testosterone serve as bimodal modulators of seizure susceptibility, demonstrating, under specific conditions, anticonvulsant as well as pro-convulsant properties. The data produced by Reddy [180] postulated that the metabolization of testosterone to estrogen increases seizure susceptibility in both animal and human models. In contrast, testosterone, through conversion to androstanediol, a positive allosteric modulator of GABAA [181,182], activated GABAA receptors, inducing chloride influx into the cell and leading to the hyperpolarization of the neuron, which can prevent neuronal hyperexcitability and diminish epileptic seizures.

Together, the above results emphasize the similarities and differences among hormone functions in epileptic conditions. Whether the action is beneficial or detrimental is still a matter of debate. The most important finding is that the antiseizure action of neurosteroids presents sex differences [172]. Future development of this research is needed for the further recognition of sex differences in epilepsies to design personalized therapies.

Despite the postulated key role of hormones, we cannot rule out the possibility of other potential factors that may determine the sex-mediated differences in epilepsy. Among the factors participating in this phenomenon are neurogenesis, glial response, chloride homeostasis, and neurotrophic factors [164]. Despite the fact that they likely play a role in determining sex differences, till now, there has been limited direct evidence of the precise functional role of these agents in models of epilepsy.

Importantly, future work is needed to understand sex differences for the design of effective therapies for epilepsy and disease modification of epileptogenesis.

## 9. Schizophrenia

Schizophrenia (SZ) is one of the most complex and least understood psychiatric disorders, showing a wide range of inter-individual variation in many aspects of the disease. It is typically referred to as a chronic and debilitating condition because it may lead to a progressive functional decline impacting cognitive, affective, and social domains [183]. The heterogenic symptoms of the disease range from hallucinations and delusions, disorganized speech and behavior, flat affect, lack of motivation, and cognitive deficits [184,185]. Cognitive impairments exist in almost all patients with schizophrenia and may determine the functional outcome of the disorder [186]. To date, the exact mechanism underlying schizophrenia is not clear.

Schizophrenia affects up to 1% of the world’s population. SZ commonly develops during adolescence or early adulthood [187]. The incidence of schizophrenia is generally higher in men than in women in ratio estimates ranging from approximately 1.2 to 1.5 [188,189]. The reverse of this ratio and the higher prevalence in women is seen in the second half of life (after 40 years).

Clinical observations and evidence in epidemiological surveys revealed significant sex differences in the incidence, clinical course, expression, and premorbid functions, as well as response to clinical treatment in schizophrenia patients [190,191,192]. In addition to the earlier age of onset and greater premorbidity dysfunction, men with schizophrenia experience more severe negative and cognitive symptoms (social withdrawal, lack of motivation, poverty of speech) relative to women [42,43,44]. Women, on the other hand, display more affective symptoms, including depression, impulsivity, self-harm, and suicide attempts [41]. The diversity most probably arises from the interplay of sex hormones and neurodevelopmental and psychosocial sex differences, but the identification of origin continues to be controversial.

The early studies tried to link sex-mediated differences to diverse structural and functional brain abnormalities in schizophrenia patients. Finally, the performed analysis yielded inconsistent results. It was found that the disparities occur in areas that normally show sexual dimorphism, implying that the same factors are engaged in both normal neurodevelopmental processes and those associated with schizophrenia. Thus, the gathering of data sets allowed us to state that men’s brains present more severe morphological abnormalities than women, including reduced frontal and temporal volumes [41,188,193,194,195].

Further systematic studies have revealed a strong connection between the levels of sex hormones and the pathophysiology of schizophrenia [36,196,197,198,199]. Importantly, it has been convincingly shown that patients with schizophrenia benefit from hormone treatments such as estrogen, progesterone, and testosterone [200,201]. The protective effect of reproductive hormones was also observed in the clinical symptoms characterized by later onset and fewer symptoms in women [200,201,202]. A significant piece of evidence concerning the action of estrogen came from the observation that during low estrogenic phases in women, greater psychosis onset and more severe symptoms were noted. The administration of hormones to postmenopausal women reduced the risk of disease, as was already noted in other impairments [198,203]. In addition, it was detected that high levels of estrogen during pregnancy may prevent psychotic relapse. Interestingly, the negative correlation between plasma estrogen levels and schizophrenia symptoms was also reported in male patients [204]. Thus, the action of estragon has received considerable attention as a probable therapeutic agent.

It has also been reported that estrogen plays a neuroprotective role by influencing major neurotransmitter systems relevant to schizophrenia as a dopaminergic system. It is known that action on the dopaminergic system is mediated by both non-genomic activity expressed stimulation of specific signaling pathways and genomic effects involving gene transcription [205]. It is hypothesized that the effect of estrogen on dopamine signaling pathways is linked to reduced impulsivity and risk of substance abuse in women [37,198]. Moreover, estragon may modify the effects of serotonergic [38,206,207] or glutamatergic systems [38,39,40] in animal studies. Estrogen increased the density of 5HT2a receptors in specific areas of the brain.

In contrast to several pieces of data concerning estragon, there is only a small amount of information available describing the effect of progesterone in the pathophysiology of schizophrenia. Halari et al. reported an association between higher circulating progesterone levels and poorer performance, specifically in spatial memory [208]. Some of the data showed that this reaction may be implicated in cognitive and emotional processing. Summarizing this discussion, it became clear that existing observations on estrogen and progesterone in schizophrenia are inconsistent, and some studies have also mentioned that these hormones have no significant effect on schizophrenic patients [209].

Another set of informative data came from analyzing the effect of testosterone. It showed that a low level of testosterone (the main male hormone) is probably associated with more severe symptoms, although results are still not completely consistent [210,211,212]. Testosterone receptors, similarly to estrogen receptors, have been implicated in the modulation of mood states and cognitive performance [213,214]. More recent reports have demonstrated that the plasma level of testosterone was inversely correlated with the severity of negative symptoms in male patients. The authors of [196,211,215] found diminished levels of testosterone in male patients and enhanced levels in female patients. These elevated levels in females correlated with brain activation during mental rotation.

Another important hormone in terms of reproductive function, oxytocin, was considered a potential therapeutic target for schizophrenia. In agreement with this suggestion, patients presenting with high levels of plasma oxytocin developed fewer psychotic symptoms [216,217] and improved cognition [218]. The inhibition of the antipsychotic effect may be related to its ability to act as a dopamine regulator [219].

A limited number of studies have been performed to identify the genetic variant that differentiates males and females [220]. So, a gender-specific association of certain dopaminergic genes, such as catechol-o-methyltransferase (*COMT*) and mono-amino-oxidase (*MAO*), with schizophrenia was suggested. Further detailed research indicated that sex-specific DNA methylation of the MAO-promoter has been associated with schizophrenia [221]. Furthermore, sex differences have also been found in GABA-ergic genes [222]. The data obtained by using GWAS found that a gene variant that encodes the zinc finger protein responsible for the regulation of dendritic spine maintenance and implicated in cognitive performance [223,224,225] was related to schizophrenia, specifically in women [226]. Similarly, the *SLC30A3* gene (which encodes protein zinc transporter 3) increases the risk of schizophrenia, again, specifically in women [227].

It is concluded that sex may affect the overall clinical picture of schizophrenia. Based on the data taken from the literature, it is suggested that the interplay of chromosomal factors and gonadal hormones may contribute to sex differences. However, attention should be paid to several additional factors that may play a role in sexual differentiation without the involvement of hormones [228]. These include differences in epigenetic mechanisms, different severity, and perinatal complications as well as exposure to viruses and environmental toxins, which could affect men and women presenting with different predispositions to schizophrenia.

## 10. Stress-Related Depressive Disorders

Stress-related disorders such as major depressive disorder (MDD) and post-traumatic stress disorder (PTSD) constitute some of the most common disabling diseases worldwide, with an approximately two-fold greater prevalence in women than in men [49,151,229,230,231]. The disorders are characterized by depressed mood, diminished interest in all daily activities, a decline in cognitive ability, and vegetative symptoms, such as disturbed sleep and appetite. They may also cause secondary disability, as patients with depression are more likely to develop chronic medical illnesses. Conventionally, depressive symptoms have been attributed to an imbalance in the hypothalamic–pituitary–adrenal axis following extreme stress created during trauma [232,233], as well as to dysregulation in the neuromodulation of neurotransmitters. Depressive disorders are also associated with inflammation [45,46].

Comprehensive studies have shown that MDD is ranked at the top of neuropsychiatric impairments as a leading cause of mortality [234]. MDD is associated with alterations in regional brain volumes, particularly the hippocampus, and with functional changes in brain circuits, such as cognitive control networks. The etiology of MDD is multifactorial, and its heritability is probably about 35% [49]. The evidence has pointed to environmental factors, such as sexual, physical or emotional abuse during childhood, which are strongly associated with the risk of developing MDD.

Epidemiological studies of MDD and PTSD found sex-based differences in the rate of disorder presentation, with an increased risk of depressive episodes in women. They may be related to the lower threshold for the manifestation of disorders. For example, it was observed that women present different initial reactivity to trauma with increased peri-traumatic dissociation and a greater negative recall, which suggests greater negative memory consolidation following stress [47]. It is worth pointing out that the progression rate of depression in females compared to males is directly proportional to the development of female reproductive function during puberty [235,236]. It indicates the interconnection of sex hormone levels and their receptors with sex differences.

In recent decades, considerable attention has been received by gonadal hormones as important mediators of sex differences between men and women [233,237]. This hypothesis is in agreement with the results of a meta-analysis that evidenced the influence of hormones on stress reactivity. Moreover, this group of researchers identified estradiol as a possible related risk factor in the prevalence and severity of PTSD between sexes. They discovered that estrogen-response elements on the pituitary adenylate cyclase-activating peptide receptor gene, *PACIRl*, determined severity in women but not men [48].

According to the previous data set, estradiol is an important regulator of the recall of extinct memories [238], which is the essential process underlying recovery from a traumatic experience. Women with higher levels of endogenous estradiol more successfully extinguish traumatic memories compared to women using contraceptives [239,240].

Summarizing the available data, PTSD and MDD disorders display clear differences between males and females. Management primarily comprises pharmacological treatment. However, there is still very little evidence of sex-related differences in terms of the efficacy of antidepressant drug treatments despite all the research performed in relation to the topic. Since both pharmacokinetics and pharmacodynamics may differ between males and females, it may become necessary to consider sex in clinical trials and provide adequately stratified guidelines. To improve therapeutic approaches, it is essential to understand the underlying mechanisms that cause males and females to respond differently to clinical interventions. Therefore, the major efforts of researchers should be directed at fully identifying the factors related to these sex-based phenomena.

An important set of supporting information came from studies on animal models including males and females. The results of several studies have revealed sex differences and the sexually dimorphic effect of some factors’ actions [241,242]. Valentino et al. have evidenced that there are sex differences in trafficking and signaling of the main modulator of the stress response-the corticotropin-releasing factor (CRF1) receptor. Sex-biased association of the CRF1 with the Gs-protein and beta-arrestin 2 would render females more sensitive to acute stress and less capable of adapting to chronic stress [241]. Moreover, Torrisi et al. discovered that the same stress-induced memory impairment in both sexes can be triggered by different sex-dependent molecular mechanisms [243]. The results of this work suggest a possible elevated NMDA and AMPA receptor sensitivity to the acute stress-induced aberrant and prolonged glutamate release in the dorsal hippocampus of female mice. Some, but not all data could be extrapolated to humans.

## 11. Conclusions

The clinical evidence reviewed here clearly illustrates pronounced sex differences in some aspects of neuropsychiatric disturbances. What emerges from this review is that much needs to be done. In particular, detailed investigations should be directed at identifying the mechanisms responsible for the sex-based difference in impairments. For now, it is still too early to translate the preliminary results obtained to routine clinical use. Nevertheless, although many questions remain to be answered, this topic may lead to avenues for novel sex-specific strategies and therapeutic options in men and women suffering from neuropsychiatric disturbances.

## Figures and Tables

**Figure 1 ijms-24-14571-f001:**
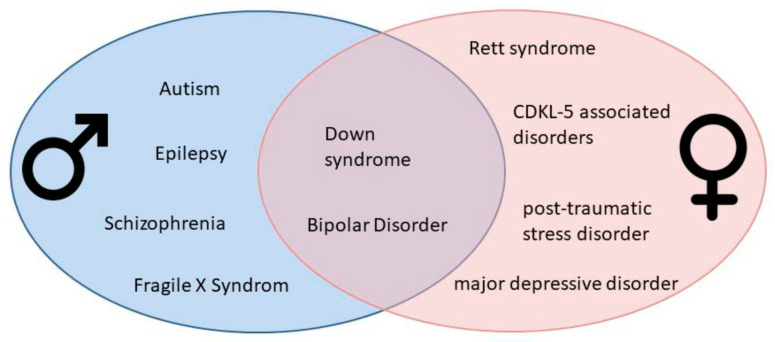
Gender bias in neuropsychiatric disorders.

**Table 1 ijms-24-14571-t001:** Sex-related differences in neuropsychiatric disorders.

	Etiology/Molecular Mechanism	Female	Male
Autism spectrum disorder (ASD)	Prenatal and perinatal altered brain development and neural reorganization (e.g., abnormal prenatal exposure to testosterone correlates positively with several ASD features) [1,2]	Less restricted interest and activities and more significant social communication difficulties [3,4,5].	More severely affected [6,7,8,9].
Down syndrome (DS)	Genetic-trisomy of chromosome 21	Higher risk of developing Alzheimer-like dementia [10,11].	Greater behavioral problems (attention deficit, hyperactivity-like behaviors in childhood, and depression in adulthood) [12] Higher prevalence of sleep disorder breathing [13]
Rett syndrome (RTT)	Genetic-mutations in the chromosome X-linked methyl-CpG-binding protein 2 (*MECP2*) gene	Intellectual disability, neurological regression, loss of speech, learning deficits, epilepsy (60–80% of RTT cases) [14]	RTT is rare (boys generally do not survive past infancy) [15,16,17,18,19]
Fragile X Syndrome (FXS)	Genetic->200 repeats of the CGG motif in the chromosome X-linked FMR1 gene and loss of its product, fragile X mental retardation 1 protein (FMRP)	Less severe symptoms than males (due to the compensation by the X chromosome) [20] Mild intellectual disability or normal intelligence [21,22]	Intellectual impairment and autistic-like behaviors [23]
Bipolar Disorder (BD)	Genetic, neuroanatomic, neurochemical and environmental factors E.g., dysregulation of several genes and pathways associated with ubiquitin pathway, synaptic functions [24], intracellular protein trafficking, protein metabolic processes [25] and neuroinflammation [26]	A higher rate of rapid cycles [27] higher number of depressive episodes [28,29] more frequent suicide attempts [30,31]	A higher rate of manic episodes and unipolar mania [28]
Epilepsy	Dysfunction of the normal connectivity of neural networks or/and imbalance of inhibitory and excitatory neurotransmitters (genetic, structural, infectious, metabolic, and immune factors)	Less frequent incidence of epilepsy than males [32,33,34,35]	Higher rate of excitability episodes and the occurrence of epileptic seizures [32,33,34,35].
Schizophrenia (SZ)	Structural and functional brain abnormalities, neurochemical disturbance (in dopamine and NMDA receptor function) and genetic factors. Estrogen has a neuroprotective effect on neurotransmitter systems: dopaminergic [36,37] and glutamatergic [38,39,40].	More affective symptoms (depression, impulsivity, self-harm, and suicide attempts) [41]	More severe negative and cognitive symptoms (social withdrawal, lack of motivation, poverty of speech) [42,43,44]
Stress-related depressive disorder (MDD, PTSD)	deregulation of neurotransmitters; neuroinflammation, imbalance in the hypothalamic–pituitary–adrenal axis after extreme stress [45,46]	Lower threshold for manifestation of depressive disorders [45,47] estradiol is a possible risk factor in the prevalence and severity of depressive disorder [48]	Lower risk of depressive episodes in men than in women [49]

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
