# Peer review of "Sex Differences in Brain Disorders"

_ijms, 2023, doi:10.3390/ijms241914571_

Round 1

Reviewer 1 Report (New Reviewer)

In this review, the authors provided up-to-date data on sex differences observed in brain disorders. It is well written, presented, and discussed appropriately. Therefore, the manuscript can be accepted in its present form.

Author Response

We would like to thank the Reviewer for the revision and acceptance of the present form of the manuscript entitled „Sex differences in brain disorders”. We appreciate your time and effort in reviewing this publication.

Reviewer 2 Report (New Reviewer)

I have read the review article on sex differences in brain disorders with great enthusiasm.  The review is very timely and thorough in terms of its presentation of the sexual divergence of a variety of neuropsychiatric and neurological conditions.  I honestly have only one major comment as I found the report to be very well-written and well-cited.

1. I would change the opening sentence of the abstract and the Introduction as most species, not just humans, show sexual dimorphic brains. Thus, humans are not that "remarkable".

There are a few awkward sentences throughout the report that could be improved in terms of the organization of phrases. English is not the native language of the authors but they did quite a good job.  

Author Response

First of all, we would like to thank the Reviewer very much for the detailed revision of our manuscript. We appreciate your time and effort in reviewing this publication.  We agree with the Reviewer, that sexual dimorphism is not only a feature of the human brain, but also many other species, so according to the Reviewer's suggestion, we have removed the phrase “human” from the first sentence of the Abstract and the Introduction.

This manuscript is a resubmission of an earlier submission. The following is a list of the peer review reports and author responses from that submission.

Round 1

Reviewer 1 Report

In this review, Ziemka-Nalecz and colleagues discuss sex differences in neuropsychiatric disorders. The manuscript is interesting. However, some points should be addressed.

-       The title is questionable. It would be better to write sex differences instead of sexual differences. Moreover, It would be better to write brain disorders instead of neuropsychiatric disorders because the Authors included also neurological disorders such as epilepsy.

-       It is quite strange that the Authors did not discuss evidence suggesting clear sex differences in stress-related disorders such as MDD and PTSD. The Authors must discuss this.

-       The Authors need to add a part discussing the therapeutic implications and the importance of conducting studies including both male and female rodents. Indeed, Several studies suggest a sex-dependent functioning of many druggable targets (PMID: 28219988; PMID: 23239826; PMID: 37293561).

Minor editing of English language.

Author Response

First of all, we would like to thank the Reviewer very much for the detailed revision of our manuscript. We really appreciate your time and effort in reviewing this publication.  According to their advice, we made some corrections and improvements to the manuscript. We hope that the Reviewer will accept the present version of our manuscript.

In this review, Ziemka-Nalecz and colleagues discuss sex differences in neuropsychiatric disorders. The manuscript is interesting. However, some points should be addressed.

-       The title is questionable. It would be better to write sex differences instead of sexual differences. Moreover, It would be better to write brain disorders instead of neuropsychiatric disorders because the Authors included also neurological disorders such as epilepsy.

Answer: We modified the title of the manuscript according to the Reviewer's suggestion. Now the title is “Sex differences in brain disorders”.

-       It is quite strange that the Authors did not discuss evidence suggesting clear sex differences in stress-related disorders such as MDD and PTSD. The Authors must discuss this.

-       The Authors need to add a part discussing the therapeutic implications and the importance of conducting studies including both male and female rodents. Indeed, Several studies suggest a sex-dependent functioning of many druggable targets (PMID: 28219988; PMID: 23239826; PMID: 37293561).

Answer: According to the Reviewer's advice we included data related to MDD and PTSD in the present version of the manuscript. We also included some sentences relating to the therapeutic implications and a set of information describing the sex-dependent effects of some drugs. We hope that the present version of the manuscript will satisfy the Reviewer. 

Reviewer 2 Report

The present paper "Sexual differences in neuropsychiatric disorders" provided an overview with up-to-date informationo in more recent findings on sex-related differences in neuropsychiatric diseases.

The structure and content of the manuscript generally meet the above-mentioned objective. I would only suggest adding a figure where some of the cellular mechanisms mentioned in the text are exemplified in a more instructive way. 

I have only found minor typos in the manuscript and spaces between words that should not be there. 

Author Response

Answer: First of all we would like to thank the Reviewer very much for the detailed revision of our manuscript. We really appreciate your time and effort in reviewing this publication.  According to your advice, we made some corrections and improvements to the manuscript. We hope that the Reviewer will accept the present version of our manuscript. According to the Reviewer's suggestion, we included a set of information concerning etiology/molecular mechanisms of describing disorders in Table 1.

Reviewer 3 Report

Already at the beginning of the manuscript there are serious problems with citations. The introduction does not contain any citations but refers to the table 1. Careful inspection of this table indicates that there are both lacking and inappropriate citations.

1) For example, the table states that for the Down syndrome (DS) there is a higher risk for developing Alzheimer-like dementia in females [30–32].

However, the reference number 30:  (Fiest, K.M.; Fisk, J.D.; Patten, S.B.; Tremlett, H.; Wolfson, C.; Warren, S.; McKay, K.A.; Berrigan, L.I.; Marrie, R.A.; for the CIHR Team in the Epidemiology and Impact of Comorbidity on Multiple Sclerosis (ECoMS) Fatigue and Comorbidities in Multiple Sclerosis. International Journal of MS Care 2016, 18, 96–104, doi:10.7224/1537-2073.2015-584 070. 585) https://www.ncbi.nlm.nih.gov/pmc/articles/PMC4849402/

is about  multiple sclerosis and does not even contain words such as “Down syndrome”, “Down's”, “trisomy”, “Alzheimer” or “dementia”.

In turn, the reference number 32: (Podcasy, J.L.; Epperson, C.N. Considering Sex and Gender in Alzheimer Disease and Other Dementias. Dialogues in Clinical Neuroscience 2016, 18, 437–446, doi:10.31887/DCNS.2016.18.4/cepperson.) https://www.ncbi.nlm.nih.gov/pmc/articles/PMC5286729/

is about dementias but does not contain any information about Down syndrome.

2) The table provides information that there is higher prevalence of sleep disorder breading (breathing?) [27,28] in men with the Down syndrome.

 However, the reference number 27: (Freeman, S.B.; Bean, L.H.; Allen, E.G.; Tinker, S.W.; Locke, A.E.; Druschel, C.; Hobbs, C.A.; Romitti, P.A.; Royle, M.H.; Torfs, C.P.; et al. Ethnicity, Sex, and the Incidence of Congenital Heart Defects: A Report from the National Down Syndrome Project. Genet Med 2008, 10, 173–180, doi:10.1097/GIM.0b013e3181634867.)

https://www.gimjournal.org/action/showPdf?pii=S1098-3600%2821%2902177-8

does not contain any information about sleep and breathing.

3) Information about Rett syndrome (RTT) in table 1 are provided without any reference. Similar situation is in case of epilepsy in females.

 Providing accurate citations is a responsibility of the authors and, therefore, the authors should carefully check the manuscript before submitting it to the journal. Serious concerns about citations mean that the manuscript is not suitable for publication and checking the content of each cited paper is not the responsibility of the reviewers.

The manuscript requires careful checking for spelling of words.

Author Response

Answer: Thank you very much for the revision of the manuscript. We really apologize for the serious errors in our references list. We carefully checked the list of references and corrected it.  We replaced the wrong citations with the corrected ones:

Fiest, K.M.; Fisk, J.D.; Patten, S.B.; Tremlett, H.; Wolfson, C.; Warren, S.; McKay, K.A.; Berrigan, L.I.; Marrie, R.A.; for the CIHR Team in the Epidemiology and Impact of Comorbidity on Multiple Sclerosis (ECoMS) Fatigue and Comorbidities in Multiple Sclerosis. International Journal of MS Care 2016, 18, 96–104, doi:10.7224/1537-2073.2015-070.

Has been changed to:

Fiest, K.M.; Roberts, J.I.; Maxwell, C.J.; Hogan, D.B.; Smith, E.E.; Frolkis, A.; Cohen, A.; Kirk, A.; Pearson, D.; Pringsheim, T.; et al. The Prevalence and Incidence of Dementia Due to Alzheimer’s Disease: A Systematic Review and Meta-Analysis. Can J Neurol Sci 2016, 43 Suppl 1, S51-82, doi:10.1017/cjn.2016.36.

We also deleted the reference “Podcasy, J.L.; Epperson, C.N. Considering Sex and Gender in Alzheimer Disease and Other Dementias. Dialogues in Clinical Neuroscience 2016, 18, 437–446, doi:10.31887/DCNS.2016.18.4” as it does not contain any information related to Down syndrome.

Moreover, we removed the reference “Freeman, S.B.; Bean, L.H.; Allen, E.G.; Tinker, S.W.; Locke, A.E.; Druschel, C.; Hobbs, C.A.; Romitti, P.A.; Royle, M.H.; Torfs, C.P.; et al. Ethnicity, Sex, and the Incidence of Congenital Heart Defects: A Report from the National Down Syndrome Project. Genet Med 2008, 10, 173–180, doi:10.1097/GIM.0b013e3181634867”. Indeed, in this publication the authors did not use the phrases “sleep” or “breathing”, however, they included the table showing the population of congenital heart defects in Down syndrome. Due to the fact that heart defect is connected with breathing disorder, we include this publication in the previous version of the manuscript. However, because there is no data related to sex-based differences, we decided to delete this position in the present version of the manuscript.

We have also completed missed references in Table 1. 

Round 2

Reviewer 1 Report

The Authors addressed all the points I raised.

Minor editing